# The Support Priorities of Older Carers of People Living with Dementia: A Nominal Group Technique Study

**DOI:** 10.3390/healthcare11141998

**Published:** 2023-07-11

**Authors:** Daniel Herron, Jessica Runacres

**Affiliations:** 1Department of Psychology, Staffordshire University, Stoke-on-Trent ST4 2DF, UK; 2Department of Midwifery and Allied Health, Staffordshire University, Stafford ST18 0YB, UK; jessica.runacres@staffs.ac.uk

**Keywords:** dementia, carer, nominal group technique

## Abstract

The aim of this study was to understand the support priorities of older (65+ years old) carers of people living with dementia. Two nominal group technique focus groups were carried out with older carers of people living with dementia. Twelve carers participated across two focus groups. Participants individually identified support priorities, and through several steps, reached a consensus to produce a ranked list of support priorities. The results consisted of two lists (one list per group), which when combined made up 15 support priorities. These priorities are presented alongside their overall and mean ranking. The authors did not refine these priorities after the focus groups, however, as there was overlap between priorities across the two liststhe results benefited from being themed. These overarching themes consisted of prioritising the carers’ holistic needs; having a sense of belonging; support needs to be accessible and timely; support to meet the wellbeing and personhood of the person living with dementia; and understanding and training for the wider community. These results have highlighted support priorities, developed by older carers, that services and organisations can use to better inform the support and services that older carers receive.

## 1. Introduction

In the UK, there are approximately 700,000 unpaid carers who act as the bedrock of support for people living with dementia [1,2]. In this article, the term ‘carers’ is used to describe unpaid informal caregivers who support a family member or friend living with dementia; this may be as co-habitants or in the residence of the person living with dementia. These carers collectively provide 1.3 billion hours of unpaid care a year [1] and, in the process, save the UK economy GBP 13.9 billion [3]. Carers spend a significant amount of their time providing essential holistic support, with 36% of carers reporting that they provide over 100 hours of care per week [4]. Caregiving is likely to come with a range of positive and negative experiences which impact upon the carer’s own quality of life [5]. For example, carers of people living with dementia have described gaining a sense of achievement and meaningfulness through providing care [6], whilst others may see it as an opportunity to strengthen their relationship with the person living with dementia [5]. However, the increased dependence of the person living with dementia, prolonged caregiving, and the progressively worsening symptoms of dementia can cause significant strain on the carer’s own physical and mental health [7]. This can be further compounded by the deteriorating physical health of both the person living with dementia and the carer, who may experience increased morbidity [2,8].

Carers have reported a multitude of factors which result in a negative appraisal of their experiences. Lindeza et al. (2020) [5], through a systematic literature review, explored the influence of dementia caregiving on family carers. They found that carers reported experiencing negative emotions, such as stress, sadness, worry, and fear, associated with, for example, making increasingly difficult decisions for the care of the person living with dementia and from seeing their declining abilities. The significant time commitment in their caregiving role also negatively impacts on carers’ ability to maintain their professional and social life and increases social isolation. A further potential consequence of their caregiving role is decreased financial wellbeing. As two-thirds of the annual cost of dementia is covered by the person living with dementia and/or their families [9], and 32% of the family’s annual household income is used for out-of-pocket care-related expenses in the last five years of the life of the person living with dementia [10], it is unsurprising that carers report increasing concern about the cost of dementia [5].

Carers’ quality of life can also be negatively impacted upon by negative social attitudes towards dementia. Bhatt et al. (2022) [11] reported that of the 70 family carers who participated in their research, 71.4% had experienced others treating them negatively when with the person living with dementia, 68.6% were excluded from family and social events, and 68.5% felt that friends avoided them. Furthermore, 78.6% of carers reported others stopped visiting their home.

The support received or not received by carers can be a mediating factor in their experiences. Some evidence has illustrated that well-supported carers provide better care for the person living with dementia and experience better outcomes for themselves [12,13]. Formal services, such as those provided by General Practitioners (GPs), the UK National Health Service (NHS), and social services, can provide essential support to meet the carers’ needs [14,15]. However, social support, such as that provided through peer support, has often been highlighted as key for promoting subjective wellbeing [15,16,17]. Herron et al. (2023) [15] reported that the remote support of family and friends during COVID-19 was essential in the absence of being able to physically meet others and attend peer support groups. Other research has highlighted the positive influence of peer support groups. Harding et al. (2023) [16] identified several support categories and corresponding behaviours which took place within support groups and highlighted, amongst other benefits, how these groups helped to develop a sense of community, connection with others, and togetherness for carers.

Though an increasing amount of literature has explored carers’ experiences of supports, only one other research study, to the authors’ knowledge, has explored the support priorities of carers of people living with dementia. Teahan et al. (2021) [2] used nominal group technique (NGT) focus groups to identify the challenges that family carers in Ireland experienced and their preferences for a range of psychosocial supports and services which addressed these challenges. Family carer participants, from groups at multiple sites, identified several carer priorities across supports and services and personal and social support. A key part of the NGT process is having participants ranking the importance of these supports and services. Results were split between personal and social supports. For personal, supports and services which provided participants with a break (e.g., day care) were ranked the highest, followed by those which helped to address challenges with social isolation. For social, supports and services which provided financial guidance (e.g., carers allowance) were ranked the highest, followed by guidance around rights and entitlements.

Given the emphasis on ensuring carers are adequately supported, and the benefits of this including maintaining and enhancing their wellbeing, it is important to gain a greater understanding of the kind of support carers want services and organisations to prioritise. Our article aimed to identify the support priorities of older carers of people living with dementia. It focuses on data collected through the NGT, like Teahan et al. (2021) [2], as this has been demonstrated to be effective in identifying support priorities; however, within our study, we have focused exclusively on older carers. This NGT was one method of data collection used as part of a larger study which utilised a multi-method approach, with carers participating in one semi-structured focus group and one NGT focus group each.

Older carers, those aged 65 years or older, were the main focus because, since 2001, there has been rapid growth in the group with a 35% increase in numbers compared to an 11% increase in carers generally over the same period [18]. Furthermore, older carers typically provide more intensive support over more hours [19]. Through a literature review, Greenwood and Smith (2016) [20] found that very little research has focused on older carers, which makes it difficult to understand their support priorities when caring for someone living with dementia.

## 2. Materials and Methods

### 2.1. Design

The COREQ Checklist has been used to report this qualitative study [21]. NGT focus groups were used to understand the support priorities of carers. The NGT is a structured approach to focus groups which allows “participants to determine which issues require further, more in-depth inquiry and to draw attention to issues that may have been previously unidentified” (Olsen, 2019, (p. 2, [22])). This method’s steps encouraged each participant to share their views through a combination of idea generation, discussion, and ranking and rating of ideas [2]. To the researchers’ knowledge, no other research has used NGTs to understand the broad support priorities of older family carers (65 years or older) within England.

### 2.2. Participants

All participants were recruited from across the West Midlands, United Kingdom, during June 2022. The inclusion criteria stated that participants had to be 65 years old or over, live in the UK, and currently be an unpaid carer of someone living with dementia. Those who did not speak English or were unable to provide informed consent were excluded.

A total of 12 participants, consisting of nine female and three male participants, with an age range of 69–81 years old, were recruited through purposive sampling. All participants were spouses to, and lived with, the person living with dementia. All participants lived alone with the person living with dementia. Participants were recruited through collaboration with a local dementia care charity who allowed both authors to attend support groups to advertise the research. This local dementia care charity provides information and support, including support and activity groups, for carers and people living with dementia. Each participant was provided with an information sheet and consent form. Participants had to sign the consent form on the day of the NGT focus groups to be able to take part in this research study. Participants were provided with a GBP 40 Love2Shop voucher for their participation in the study. See Table 1 for demographic details.

### 2.3. Data Collection

Each participant took part in one NGT focus group. Following guidance from the literature, the number of participants was restricted to a maximum of 7 per NGT focus group [23]. Participants had the option to select one of two dates to attend; therefore, the participants were split into two groups (group A = 5 participants; group B = 7 participants). Both days of data collection were in person and took place in a conference room of the local dementia care charity. To facilitate participation and inclusivity, it was agreed that the carer could bring the person they provided care for, and that the local dementia care charity would provide a separate, staffed room of activities. Having a space for just carers better allowed them to speak openly. Refreshments, snacks, and lunch were provided for all participants and the person living with dementia. Regular breaks were provided throughout the day. NGT focus groups were audio-recorded and transcribed verbatim by the authors, with permission from the participants. Both authors facilitated the NGTs and followed procedures shared by Gaskin (2003) [24] and Olsen (2019) [22] (see Table 2). NGT focus group A lasted 68 minutes and group B lasted 65 minutes.

### 2.4. Ethics

Ethical approval was provided by Staffordshire University Ethics Committee (SU_21_151). All participants were made aware that participation was voluntary and provided written and verbal informed consent prior to participating in data collection. Participant names have been replaced with pseudonyms and identifying information removed.

### 2.5. Analysis

Two lists of items were developed, one from each group, and these provided the results from the NGT focus groups. To ensure that the support priorities identified by the carers were at the forefront of our results, the authors did not refine these lists any further after the NGT focus groups. However, both authors agreed that there was overlap between items across the two lists and the results would benefit from theming the items identified by carers. Analysis started with both authors reading each list of items and the meaning of each item and making initial notes of anything of interest. Both authors independently initially themed the data from the two lists. They then met to review each other’s themes and collaboratively developed a shared list of themes. This new list of themes was reviewed by the authors until they were satisfied that themes were underpinned by a central concept, captured the data, and addressed the research question. Any disagreements across the themes were discussed between the authors during analysis meetings, with the outcome agreed upon by consensus. The final list of themes was shared and discussed with the local dementia care charity, and any feedback was used in the development of the final list of themes.

## 3. Results

In total, 15 items were developed across the two NGT groups (group A = 8; group B = 7). Table 3 (group A) and Table 4 (group B) provides the list of ranked support priorities which were developed and agreed upon by carers, alongside their overall score and mean score.

Table 5 provides a list of the four themes developed by the authors and states which items from group A’s and B’s lists have informed the theme. These items and their meaning are described under a relevant theme heading.

### 3.1. Prioritising the Carers’ Holistic Needs

Ranking highly on both lists of priorities were list items which focused on prioritising the holistic needs of carers. These holistic needs included maintaining their interests and hobbies and support for their own mental and physical health. Carers discussed the need for support from services to prioritise their holistic needs alongside that of the person living with dementia:
“…what happens if I fell, your illness is second to that person (and it should be first), and it should be yes, cause you need be able to take care of them”.(Carer Group B)

The deteriorating health of the person living with dementia and associated increasing care responsibilities meant that many of the carers felt they had less time to manage their own increasing health concerns and needs, which was compounded by their own health needs being put second by social and healthcare services. Consequently, carers spoke about their worry of not being able to provide the necessary support and care for the immediate and future needs of the person living with dementia.

Carers also felt that taking on these increasing care responsibilities had impacted upon their personal identity. In many cases, carers had no choice but to stop engaging in many of the things they enjoyed and which were important for them:
“But I can only do it [attend dance events] when [partner living with dementia] goes in respite. I can’t do it … and yet I have been involved for 44 years, that’s how long I have been involved. And I used to go three times a week…”.(Carer Group B)

To alleviate this, carers discussed prioritising support which provided opportunities to continue in engaging in things they enjoy.

A common theme across both NGT focus groups was the need to have time away from their caring responsibilities so the carer could prioritise their own holistic needs. Carers made several suggestions on how services could facilitate this. For example, there was a consensus that there needed to be more support for accessing respite and day centre services which could facilitate time to focus on the carer’s own needs.
“My priority is respite…my partners in respite now and it does give me a rest”.(Carer Group B)

This also included having time to speak to healthcare professionals about their own health without having to attend to the needs of the person they were supporting.

### 3.2. Having a Sense of Belonging

Ryan et al. (2008, (p. 80, [26])) define in their relationship-centred care framework a sense of belonging as being “able to confide in trusted individuals to feel that you’re not ‘in this alone’”. Carers illustrated the importance of having a sense of belonging through items which emphasised access to support groups where they can share their experiences with others who may have similar experiences:
“We need groups to share our experiences”.(Carer Group A)

Carers felt frustrated with the difficulty of accessing information about support groups which were provided across local organisations and services. Subsequently, carers spoke about the need for these organisations and services to prioritise cross-organisational and service communication to raise awareness of available support groups and for information about these support groups to be collated and shared with carers.
“…I want better liaison between the different services and organisations. Nobody seems to come together…they’re not talking between one another…”.(Carer Group B)

This collated source of information would better allow carers to access support groups which they felt would support a sense of belonging with other carers.

Carers also felt it was a priority to ensure that support groups, which they had already accessed, were better suited to their need to connect with other carers and to have open discussions about their experiences. At the time of the NGT focus groups, carers attended support groups with the person living with dementia; however, they felt that this stopped them from sharing their experiences in fear of upsetting the person living with dementia. Instead, carers suggested prioritising the development of support groups which replicated that of this study’s focus group structure, where people living with dementia were being supported by staff and taking part in activities in a separate room from carers. Carers felt that they would be able to openly share their experiences with each other.
“So we can share experiences, so what I had in mind was something like today, so partners were cared for…”.(Carer Group A)

### 3.3. Support Which Is Accessible and Timely

Carers across both groups developed items which illustrated the need to prioritise support that was available to carers in a timely manner. In Group B, carers discussed their difficulties of knowing whom to contact for emergencies or for specific health concerns for the person living with dementia:
“…what happens if I need to ring someone, who do I call… what happens if they [partner living with dementia] do fall”.(Carer Group B)

Many carers shared stressful experiences of being told of their partner’s diagnosis of dementia by a healthcare professional and receiving very little information about possible supports. Carers, especially those in group B, discussed a support priority being the need for support, which could also be in the form of information to be provided at the time of a diagnosis:
“And a hell of a lot more information. Don’t you agree [a different carer]?...They presume that you know. They presume that because you are the wife, or the husband, whichever the case may be, that you know what to do”.(Carer Group B)

For carers, it was a priority that timely and accessible support was ongoing and aligned with key points in their journey supporting the person living with dementia:
“…make it obvious to a newly caring person what help is available…”.(Carer Group A)
“We need access to support and advice on practical matters i.e., wills, power of attorney etc.”.(Carer Group A)

Some participants spoke about the need for greater support and information at a time where the person they are supporting, for example, become less mobile and the need to understand what support is available and how they access this support to help them maintain the dignity of their loved ones. One carer spoke about how timely support would have been useful for them:
“help in the home…once I’ve dragged her [partner living with dementia] downstairs from bed, I’ve had it, I can’t move. I need specialist help”.(Carer Group B)

### 3.4. Support to Meet the Wellbeing and Personhood of the Person Living with Dementia

Across both groups, carers highly ranked support priorities which better enabled them to enhance the personhood and health of the person living with dementia. There was more of a focus on these support priorities in Group A (3 items) compared to Group B (1 item). Carers felt it was important that they were supported to better understand how dementia progressed, how this presented through symptoms, and what the carer could do to help alleviate these symptoms:
“…we want advice on what to expect because we don’t know how things are going to progress until it happens and then we look it up and think oh yes…”.(Carer Group A)

Carers in Group A spoke about wanting more information on what they could and should be doing on a daily basis to support the needs of the person living with dementia:
“What matters most in the caring situation. So prioritising what the carers should look at as the most important things to do”.(Carer Group A)

Carers in Group B discussed needing more support with navigating what funding and support was available and completing the associated forms to ensure the support was in place for the person living with dementia and to help them to thrive:
“We need support filling in forms and understanding what support and funding is in place…we are stumbling in the dark…”.(Carer Group B)

Other participants in Group B described trying to find relevant information as a “mine field” due to the difficulty of the task.

Both groups wanted a central source of information which they could access. For example, a carer in Group B stated the following:
“Having somewhere which provides a list of places we can go to and things we can do…like which are dementia friendly”.

Group A’s highest ranked priority was the need for support to better understand how they could meet the emotional needs of the person living with dementia:
“How much does a partner need love, friendship etc., even if they don’t communicate. How do we know they still don’t understand”.(Carer Group A)

This group’s carers discussed their challenges of understanding these needs, especially when people living with dementia had impaired communication.

### 3.5. Understanding and Training for the Wider Community

A prominent support priority for carers in Group B extended to the wider community, where there was a belief that people across organisations and services needed a greater understanding of how to promote the personhood of people living with dementia, which required the undertaking of dementia training:
“Understanding from the management, from the top all the way down to the bottom to the helpers and carers…they need training”.(Carer Group B)

This support priority was informed by the carers’ polar experiences of community understanding. Some carers discussed the understanding of the local community and healthcare services who had facilitated the personhood of the person living with dementia and eased some of the concerns of the carer when out in the community. For example, the understanding of members of one local older person group enabled one carer, with the person living with dementia, who had both been members of the group prior to the diagnosis of dementia, to continue attending and feel connected to the rest of the group.

## 4. Discussion

Carers of people living with dementia take on substantial caring responsibilities [15] which saves the UK economy GBP 13.9 billion [3]. However, their role may mean making significant sacrifices which can impact their own wellbeing [15]. It is therefore reasonable that carers should receive the necessary supports and services to support them along their caring journey and to ensure their wellbeing is also prioritised. This is particularly important since, as previously stated, well-supported carers are more likely to provide better care for the person living with dementia and experience better outcomes for themselves [12,13]. This makes it important to understand what support carers prioritise. This research study’s unique use of a NGT has been effective in providing important understanding and expands our knowledge of these support priorities for a group of carers. The NGT provided a platform for carers to directly share their support priorities.

In total, across both groups, 15 support priorities were shared which the authors themed. Unsurprisingly, many of the top-ranked priorities reflected the carers desire to ensure the person living with dementia received support which promoted a sense of personhood and the holistic wellbeing of the person living with dementia. These priorities highlighted that carers want to understand the ongoing, changing, complex needs of the person living with dementia and the complexities of navigating funding so they can better ensure appropriate care is provided. Carers further laminated the need for dementia services and organisations to support carers to access person-centred dementia training and to provide support in accessing funding. This is supported by research that has illustrated carers want for greater support to access funding and to fill out the associated forms [2] but also the importance of carers’ awareness and knowledge of dementia, how to care for people living with dementia [27,28,29], and how both can assist in building confidence in providing dementia care [30]. This support can help to alleviate the burden and anxiety experienced by carers [31].

Research has consistently shown the important role communities play in supporting the wellbeing and personhood of the person living with dementia [32]. Community is an important consideration within the development of dementia-friendly communities [33], with frameworks and guidance highlighting the need to understand and change how people, such as neighbours, health and social care professionals, and the wider community, support and respond to people living with dementia [34,35]. Quinn et al. (2022) [32], through a large questionnaire study, highlighted that carers and people living with dementia wanted greater understanding and awareness of dementia within the community and felt that this would lead to those in the community treating people with dementia better. This was supported by carers within our research study who highlighted community understanding and awareness of dementia as a priority. This research, like others [15,27,32], supports the need for public awareness initiatives, including training, to distil dementia awareness within the wider community.

Carers in our research study have highlighted the need for dementia services and organisations to prioritise their holistic needs alongside those of people living with dementia. An abundance of literature has shown the strain carers may experience and the negative impact this can have on their own physical and mental health [7,15] and has highlighted the importance of meeting the holistic needs of carers [5,13,26]. This makes it vital that carer support priorities, which target their holistic needs, are provided. Within our study, a collection of these priorities revolved around the carers wanting relief from their caring role, so they had time to engage and enjoy the things they did prior to dementia, such as their hobbies. One way of achieving this, and a support priority of carers in our research, was through respite. As a support priority, this was also reported by Teahan et al. (2021) [2], who showed that carers prioritise support, such as short-break and long-break respite, which enabled them to have regular breaks for their own wellbeing. Respite is not a one size fits all, and several factors can impact on how effective it is, such as the length of the respite [36,37]. Therefore, it is recommended that dementia services and organisations prioritise good quality respite which meets the individual needs of the carer.

Alongside relief from caring responsibilities, carers prioritised a sense of belonging [26]. Having access to, and being made aware of, support groups was raised as a support priority for carers in this study. Carers wanted opportunities to confide in and share their experiences with trusted fellow carers and to learn from their experiences, knowledge, and the resources they use. Previous research has highlighted this as a carer priority [2] and has shown that support groups are associated with good subjective wellbeing [17], a source of positive emotional wellbeing [38], and a space to share experiences and resources [17]. The carer priorities also highlighted the need for closer collaboration between organisations and services to ensure groups are regular and that carers know when they can attend them. Interestingly, carers in one group spoke about their dissatisfaction which the current set up of their support group, which consisted of the carer and the person living with dementia and meant that they were concerned about upsetting the person living with dementia. Instead, carers in one group suggested that support groups are run like our NGT focus group, which meant that people living with dementia were cared for in a separate room where they also took part in dementia-friendly activities. This would better allow carers to openly discuss their experiences without upsetting the person living with dementia. We recommend that dementia services and organisations not only more effectively communicate with each other and share information about support groups facilitated across services and organisations, but that they consider whether they can provide separate support for people living with dementia at the time of carer support groups. This would likely facilitate carers’ attendance in support groups, provide some relief from caring responsibilities, and allow them to share their experiences freely.

Importantly, carers wanted accessible and timely support prioritised across their journey with dementia. This was no more evident than during the diagnosis process, where carers discussed not being provided with any support after being told the diagnosis. At a time where carers were processing the diagnosis, they also reported having to research the diagnosis due to the lack of support on understanding dementia and what it meant for the person living with dementia. Examples such as this highlight the need for quality interactions during the diagnosis process, where timely information is provided to support carer understanding. This is supported by Woods et al. (2018) [39], who explored family carers’ experiences across five European countries. Through a mixed methods questionnaire, they illustrated an association between higher quality diagnostic disclosure and “better adjustment and less negative emotional impact on carers in the short and medium term” (p. 114, [39]). Additionally, when exploring what made high-quality diagnosis sharing with carers, statements around well-prepared healthcare professional, clear communication, opportunities to ask questions, and future planning during the diagnosis meeting were ranked highly. Many of these contrasted with what participants experienced in our research, which may explain why they generally viewed the diagnosis processes negatively.

### Limitations

There are several strengths and weakness which are important to consider for any future research using the same approach. Applying the NGT allowed this research to generate a list of support priorities which have been developed and ranked by carers themselves. This has provided a useful list of priorities which organisations and services can use to better inform the support provided to carers. To better facilitate inclusivity for carers and as recommended by Teahan et al. (2021) [2] and further discussed in Runacres and Herron (2023) [40], the authors actively discussed the financial implications of participating in this study. Participating meant a substantial time commitment (4–5 h of their day) and without the appropriate support, it could have meant carers organising and paying for replacement support for the person living with dementia. To minimise the financial implications of participation, the authors, within their funding application, factored in payment for carer time (GBP 40 Love2Shop voucher) and for the local dementia organisation in which the focus groups were taking place to provide a separate space in the building which was managed by professional carers who supported the person living with dementia. We also costed for food for both the carer and person living with dementia; this was something which was particularly valued during a cost-of-living crisis. These are important considerations for research which wishes to include diverse carer experiences [40].

There are limitations to consider when interpreting the results of this study. Future research should carefully consider the sample used within their research. Within our study, the sample provided important insight into carer support priorities. However, the sample size, whilst comparable to other NGT studies with a similar population [2], was small and all participants attended support groups provided by one local dementia care organisation. This may mean that the support priorities developed in this study do not reflect other carers who may not be able to attend support groups. For example, carers were recruited through advertisement of the study within face-to-face support groups. Yet, carers who need to find alternative support for the person living with dementia to attend peer support groups or having work commitments, may be less likely to attend in-person support groups [41,42], so they may not have had their priorities represented in this study. All participants were from one county in England, so the support priorities they have raised may be reflective of their experiences of the services or lack of services provided in that county and this may differ to carers in other counties within the UK. However, many of the support priorities within our study are shared by research with carers in Ireland [2] which supports the careful application of these results more widely. Future research needs to take steps to ensure inclusivity within their research and to try and reach as many carers as possible with diverse experiences to generate their support priorities.

## 5. Conclusions

Carers in this research have produced a wide-ranging list of support priorities which centred around ensuring that the ongoing holistic needs of the person living with dementia are met, prioritising the carer’s holistic needs, supporting their sense of belonging, ensuring that support is accessible and timely, and facilitating support to raise awareness and understanding within the wider community. Some of the highest ranked items focused on priorities which supported them to provide the necessary care for the person living with dementia, such as understanding their emotional needs and the progression of dementia. Other highly ranked items focused on support that provided relief from caring duties (e.g., respite) and promoted a sense of belonging (e.g., peer support groups). These results have provided several implications for dementia services and organisations, including providing greater support to alleviate carers of their caring responsibilities and ensuring identifiable support for carers from the diagnosis of dementia which is maintained across their journey as a carer. Carers also want their current peer support group set-up changed to allow them to freely discuss their experiences with other carers.

This research has shown that the NGT is effective in identifying the support priorities of a group of older (65+) carers of people living with dementia. The stages of the NGT produced a list of priorities which were collectively ranked and will be useful to services and organisations which support carers of people living with dementia. Future research within this area needs to consider ways to ensure that diverse caring experiences are included. Carers were positive about the processes the authors implemented to reduce the financial burden of participating, including providing on-site support for the person living with dementia.

## Figures and Tables

**Table 1 healthcare-11-01998-t001:** Participant demographics and contextual information.

NGT Group (A or B)	Participant ID	Gender	Age	Live with Person with Dementia	Gender of Person Living with Dementia	Age of Person Living with Dementia	Type of Dementia
A	Amy	Female	74	Yes	Male	85	Alzheimer’s disease
A	Sophie	Female	73	Yes	Male	81	Alzheimer’s disease
A	Maya	Female	70	Yes	Male	71	Alzheimer’s disease
A	Jenny	Female	78	Yes	Male	80	Mixed dementia
A	Sarah	Female	78	Yes	Male	78	Parkinson’s disease dementia
B	Nina	Female	79	Yes	Male	82	Alzheimer’s disease
B	Nathan	Male	78	Yes	Female	76	Alzheimer’s disease
B	Hadrian	Male	86	Yes	Female	83	Mixed dementia
B	Patricia	Female	76	Yes	Male	79	Vascular Dementia
B	Constantine	Male	69	Yes	Female	80	Alzheimer’s disease
B	Rhea	Female	70	Yes	Male	73	Parkinson’s disease dementia
B	Eve	Female	81	Yes	Male	83	Vascular dementia

**Table 2 healthcare-11-01998-t002:** Steps taken when using the nominal group technique (NGT).

Step Label	Step Procedure
Step 1. Introduction to NGT procedure and focus	At the start of each NGT, the authors shared and explained the NGT procedures and presented the research question participants considered: ‘What are your support priorities as a carer of a person living with dementia?
Step 2. Initial idea development	Carers were provided with a pen and paper and instructed to individually and silently generate ideas of their support priorities and asked to list these on a piece of paper.
Step 3. Round robin	In a round robin recording of ideas, each carer shared an item, in turn, until all carers had shared an item. This process was repeated until each carer had shared all items on their list. Carers were encouraged to avoid sharing items which they felt were the same as any already shared but to share items that they felt differed in any way. All items were simultaneously typed up by the authors and shared on a projector screen.
Step 4. Clarification of items	The researchers the encouraged the whole group of participants to discuss each item on the list to clarify the wording and meaning and to remove and merge overlapping items [25]. A completed list, which was agreed upon by all participants, was then shared on the projector screen.
Step 5. Individual ranking of items	Participants then individually and anonymously selected and ranked each of the items based on how important of a priority the item was to them. Rankings were based on a scale from 1 to 10, with 1 representing low priority and 10 representing high priority. Participants wrote their votes down on a piece of paper which was collected by the authors.
Step 6. Ranking consensus and discussion	Rankings were then tabulated by the authors and the results were shared in the group on a screen. Finally, participants discussed their thoughts of the list and item order.

**Table 3 healthcare-11-01998-t003:** NGT Group A list of support priorities.

List Ranking Position	Item	Overall Score	Mean Score	Identifying Code
1	How much does the partner still need love and friendship? Even if they do not communicate, how do we know?	47	9.4	A1
2	Education on illness progression and how to plan for the future.	44	8.8	A2
3	Communicating new treatment.	38	7.6	A3
3	Day centre service and funding to facilitate time without caring responsibilities.	38	7.6	A4
4	1:1 contact in time of stress. Knowing who best to contact? e.g., an emergency number, not the police, ambulance, etc.	36	7.2	A5
5	Group talks to share experiences. Care for person living with dementia to facilitate engagement.	33	6.6	A6
5	Accessible and timely information and support (e.g., stair lifts, care homes, wills, power of attorney).	33	6.6	A7
6	What should you prioritise in the day?	29	5.8	A8

**Table 4 healthcare-11-01998-t004:** NGT Group B list of support priorities.

List Ranking Position	Item	Overall Score	Mean Score	Identifying Code
1	Support for understanding funding (e.g., filling in forms, access, care allowance)	58	9.6	B1
1	Understanding and training for the wider community. Dementia-friendly community.	58	9.6	B2
2	Being able to talk to a healthcare professional (without the person living with dementia present)	55	9.1	B3
3	Prioritising the carer (e.g., identity, mental and physical health)	54	9	B4
3	Sharing information of support groups between services and to carers	54	9	B5
4	Information and support from professionals from the start of the process	49	8.1	B6
4	Time without caring responsibilities (e.g., respite and sitting services)	49	8.1	B7

**Table 5 healthcare-11-01998-t005:** List of themes.

Theme Name	Brief Description	Item/s Informing the Theme
Prioritising the carers’ holistic needs	Prioritising services and organisations to prioritise their holistic needs alongside that of people living with dementia	A4, B3, B4, B7
Having a sense of belonging	Prioritising support groups which provide carers with a space to openly discuss their experiences	A6, B5
Support needs to be accessible and timely	Prioritising support that is ongoing and which carers can access at a time they need it	A5, A7, B6
Support to meet the wellbeing and personhood of the person living with dementia	Prioritising supports which better enabled carers to enhance the personhood and health of the person living with dementia (e.g., information to navigate funding)	A1, A2, A3, A8, B1
Understanding and training for the wider community	Prioritise improving understanding through training for those in the wider community to alleviate care concerns of going out into the community with the person living with dementia	B2

## Data Availability

Data is available from the authors upon reasonable request.

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
