# Peer review of "The Support Priorities of Older Carers of People Living with Dementia: A Nominal Group Technique Study"

_healthcare, 2023, doi:10.3390/healthcare11141998_

Round 1

Reviewer 1 Report

The aim of the paper "The support priorities of older carers of people living with dementia: A nominal group technique study." is to understand the support priorities of older (65+ years old) carers of people living with dementia. The results have pointed support priorities, developed by older carers, which could lead to better information of services that older carers need.

1.      English language: The English language used is excellent.

2.      Title: The title is adequate.

3.      Abstract: The abstract is nicely written and provides the necessary information.

4.      Introduction: The introduction is extremely informative and understandable and gives a good background of the topic. It contains the right amount of information without overwhelming the reader. In the sentence, "Older carers, those aged 65 years or older, were focused on as they are a rapidly growing group with a 35% increase in numbers compared to an 11% increase in carers generally since 2001," it is unclear from which year there is an increase of 35% (in 2023?).

5.      Materials & Methods: This section is precise and clearly described. Is "EVA8" in Table 1 a typo? I am aware that participant names have been replaced with pseudonyms, as stated in the line 153, I am just not sure about the number 8.

6.      Results: The results are clearly presented, nicely written, and easily understandable.

7.      Discussion: The discussion is comprehensible and adequately addressed. The literature cited is relevant to the study. There are some differences in formatting of the section Discussion compared to the section Results. All my concerns and questions have been already clarified and answered in the section Limitations.

Overall, the paper titled "The support priorities of older carers of people living with dementia: A nominal group technique study" is written in an appropriate manner. It is well-structured and interesting to read, providing a new perspective on the challenges faced by carers of individuals with dementia. It is also intriguing to see how the UK is addressing these issues. It would be beneficial (not for the authors of the current manuscript, but in general) to compare how other countries are dealing with similar problems in different parts of Europe and other continents. This study is technically sound and the manuscript itself is likely to attract a wide readership. Therefore, the manuscript can be considered for publication.

Author Response

Thank you for reviewing this manuscript and your feedback. Below I have added a table which states your feedback point to address and my response. 

Reviewer 1

Response

4.      Introduction: The introduction is extremely informative and understandable and gives a good background of the topic. It contains the right amount of information without overwhelming the reader. In the sentence, "Older carers, those aged 65 years or older, were focused on as they are a rapidly growing group with a 35% increase in numbers compared to an 11% increase in carers generally since 2001," it is unclear from which year there is an increase of 35% (in 2023?).

The wording has now been changed to make it clearer from which year I am referring to.

“Older carers, those aged 65 years or older, were focussed on as since 2001 there has been  rapid growth in the group with a 35% increase in numbers compared to an 11% in carers generally in the same period [18].”

5.      Materials & Methods: This section is precise and clearly described. Is "EVA8" in Table 1 a typo? I am aware that participant names have been replaced with pseudonyms, as stated in the line 153, I am just not sure about the number 8.

I have checked the document for typos and made corrections. I have removed the 8.

Reviewer 2 Report

Introduction

Point 1: I think that a sufficient number of literature reviews have been conducted. It would be better to state the purpose and necessity of the study more clearly at the end of the introduction.

Methods

Point 2: Were there considerations of reliability and validity in this qualitative research method? What are the efforts to maintain objectivity and eliminate bias?

Point 3: Although it is a qualitative study, I am concerned about small sample size and representativeness issues. Please add the possible problems with these issues in this research in the Limitation section.

Author Response

Thank you for reviewing this manuscript and providing your feedback. Below, I have added a table which states your feedback and my response to your feedback. 

Reviewer 2

Response

Introduction

Point 1: I think that a sufficient number of literature reviews have been conducted. It would be better to state the purpose and necessity of the study more clearly at the end of the introduction.

I have more clearly stated the need and purpose of this study close to the end of the introduction, and ensured this fits within the structure of the introduction whilst maintain the flow of the introduction.

Given the emphasis on ensuring carers are adequately supported, and the benefits of this including maintaining and enhancing their wellbeing, it is important to gain a greater understanding of the kind of support carers want services and organisations to prioritise. Our article aimed to identify the support priorities of older carers of people living with dementia”

Methods

Point 2: Were there considerations of reliability and validity in this qualitative research method? What are the efforts to maintain objectivity and eliminate bias?

Point 3: Although it is a qualitative study, I am concerned about small sample size and representativeness issues. Please add the possible problems with these issues in this research in the Limitation section.

Quality criteria, such as validity, reliability and objectivity are not applicable to this qualitative study design (these concepts are more applicable to a quantitative design) and these have therefore not been considered. Instead, and as is stated in the method section of the article, we have used the COREQ Checklist to ensure transparency and quality. Through this checklist, we have provided details which better ensure the quality of the research, such as the credentials of both researchers, a clear audit trail of the theming process and what we did if there were differences in theming.

Research using the same methodology have similar sample sizes. Representativeness, as applied more within a quantitative methodology (i.e. a sample that allows statistical inference to be made to an entire population) is not generally applied to Nominal group technique. I agree that the sample size and characteristics are a possible limitation to the study and have already discussed this in the limitations section (see p14-15) whilst illustrating the possible drawbacks of the sample (including the size) and what this may mean for the transferability of the results.

Reviewer 3 Report

1) In statistic analysis, mean and standard deviation (SD) are two important factors, however, the authors didn't provide SD in the score table. In addition, there is a parenthesis in Table 3(NGT Group A List of Support Priorities), more explanation should be provided for the strange symbol.

2) Typos and grammar mistakes can be spotted in the manuscript along with the missing table's heading (e.g. Table 5). The manuscript should be carefully checked prior to the re-submission.

3) More detailed analysis should be conducted ahead of giving further conclusion. For example, is the difference between Group A and B significant? Does the ranking order of listed items illustrate any stochastic pattern?

4) Does the health status of carers influence the decision of ranking priority items? Do all the participants (older carers) live alone with their spouse? Is there any assistance from their children, does this impact on the priority selection?

Typos (e.g. focussed in Line 99) exist along with grammar mistakes. 

Author Response

Thank you for reviewing this manuscript and providing your feedback. Below, I have added a table which states your feedback and my response to your feedback. 

Reviewer 3

Response

1) In statistic analysis, mean and standard deviation (SD) are two important factors, however, the authors didn't provide SD in the score table. In addition, there is a parenthesis in Table 3(NGT Group A List of Support Priorities), more explanation should be provided for the strange symbol.

As this study has not used a quantitative design, we have not carried out in-depth statistical analysis. Means have been provided to illustrate decisions made within a key stage of the NGT, however, SD has not been provided as this would not align with the NGT methodology. We have also been guided by previous studies which have used NGT (Rice et al., 2018; Teahan et al., 2021).

The parenthesis was a typo and has now been removed.  

2) Typos and grammar mistakes can be spotted in the manuscript along with the missing table's heading (e.g. Table 5). The manuscript should be carefully checked prior to the re-submission.

I have checked over the manuscript and corrected typos. I have also included a heading for table 5.

3) More detailed analysis should be conducted ahead of giving further conclusion. For example, is the difference between Group A and B significant? Does the ranking order of listed items illustrate any stochastic pattern?

This study has not used a quantitative design and therefore, such forms of analysis would not be appropriate (or possible) with the collected data. This is inline with other studies (referenced in the manuscript; Rice et al., 2018; Teahan et al., 2021) which have focused on the qualitative element of the nominal group technique.

4) Does the health status of carers influence the decision of ranking priority items? Do all the participants (older carers) live alone with their spouse? Is there any assistance from their children, does this impact on the priority selection?

As this study has not used a quantitative design, we are unable to explore cause and effect, associations or relationships between variables. Furthermore, data was not collected on the health status of the carers, though some carers explored their health and the impact it had upon them when developing the support priorities.

I have added the following comment in the text to highlight that all participants live alone with their spouse

“All participants lived alone with the person living with dementia”

The purpose of this study was for carers to develop support priorities and we used a methodology which provides a process to achieve this; however, this focus and the nominal group technique did not aim to explore the specific area of assistance from their children and the impact this had on how they developed their priority. Therefore, I feel it is outside of the ability of this study to explore and comment on this.

Round 2

Reviewer 3 Report

It is an interesting pilot research. It would be more be beneficial for guiding the support priority if deepened study on the relationship between carer's health and dementia patients were conducted.

Author Response

Thank you for your thoughts on our manuscript. Though this paper did not focus on the relationship between carer's health and the person living with dementia, it is an interest area and one which could be explore in future research. Thank you for your suggestion.  

 It would be more be beneficial for guiding the support priority if deepened study on the relationship between carer's health and dementia patients were conducted.